# Integration of Circulating miR-31-3p and miR-196a-5p as Liquid Biopsy Markers in HPV-Negative Primary Laryngeal Squamous Cell Carcinoma

**DOI:** 10.3390/diseases13090279

**Published:** 2025-08-27

**Authors:** Gergana Stancheva, Silva Kyurkchiyan, Iglika Stancheva, Julian Rangachev, Venera Dobriyanova, Diana Popova, Radka Kaneva, Todor M Popov

**Affiliations:** 1Molecular Medicine Center, Department of Medical Chemistry and Biochemistry, Medical Faculty, Medical University of Sofia, 1431 Sofia, Bulgaria; gsstancheva@gmail.com (G.S.); kaneva64@gmail.com (R.K.); 2UMHAT “Tsaritsa Yoanna—ISUL”, Department of ENT, Medical University of Sofia, 1537 Sofia, Bulgaria; iglika_stancheva@yahoo.com (I.S.); jrangachev@abv.bg (J.R.); v.dobriyanova@mu-sofia.bg (V.D.); dp_popova@abv.bg (D.P.); popov@todorpopov.com (T.M.P.)

**Keywords:** laryngeal cancer, HPV-negative, circulating miRNAs, miR-31-3p, miR-196a-5p, liquid biopsy, plasma biomarkers

## Abstract

Background and Objectives: Laryngeal cancer is a common head and neck tumor burden, with no significant improvements in long term patient survival. Despite the progress of molecular genetics and oncology strategies, there is still a lack of biomarker use in routine clinical practice for early laryngeal cancer screening or diagnosis. miRNAs are explored as promising molecules, that could serve as liquid biopsy. Our goal is to explore the screening potential of miR-31-3p and miR-196a-5p in early- and advanced-stage laryngeal HPV-negative plasma samples. Methods: In this study, 50 plasma samples obtained from early and advanced HPV-negative laryngeal cancer patients were included. The expression levels of mir-31-3p and miR-196a-5p were analyzed via TaqMan RT-qPCR. SPSS v27.0 was used for statistical analysis. Results: For the first time, miR-31-3p and miR-196a-5p were analyzed in plasma samples from early HPV-negative primary LSCC patients. Both circulating miRNAs showed significantly elevated expression levels in early and advanced laryngeal cancer samples. miR-31-3p was significantly associated with T stages (*p* < 0.001) and N stages (*p* = 0.009). The ROC analysis revealed that miR-31-3p could significantly discriminate early-stage from advanced-stage LSCC with an AUC of 0.850 (95% CI: 0.743–0.956, *p* < 0.001) at an RQ cutoff of 2.03, achieving a sensitivity of 95.5% and a specificity of 64%. Nevertheless, miR-196a-5p was found to be significantly overexpressed in early-stage LSCC, which could contribute to the development of its screening potential. For the first time, both miRNAs revealed a significant positive correlation, which indicates that miR-31-3p and miR-196a-5p could coregulate cancerogenesis. Conclusions: In conclusion, the data revealed that miR-31-3p has greater potential as an LSCC screening marker in comparison to miR-196a-5p. Still, miR-196a-5p also showed promising results in early-stage laryngeal cancer monitoring. The utilization of circulating miR-31-3p or miR-196a-5p analysis could enable liquid biopsy approaches, with results potentially informing treatment monitoring strategies, personalized oncological protocols, and early diagnosis. These advancements could ultimately benefit patient outcomes by improving laryngeal organ preservation and survival rates.

## 1. Introduction

Primary laryngeal squamous cell carcinoma (LSCC) is the second most common type of head and neck cancer (HNC). In 2022, an estimated 40,387 cases of LSCC were reported in Europe, with 19,090 deaths (both sexes), according to GLOBCAN data. There is a significant disparity in incidence between males and females, with men being markedly more affected. In Bulgaria, the majority of patients are diagnosed at an advanced stage of LSCC, with an incidence rate of 6.0 per 100,000 and a mortality rate of 5.0 per 100,000 (GLOBCAN, 2022, males). The data indicate a very poor disease course, with worse prognoses and outcomes.

Laryngeal cancer is more common in people over the age of 55. Less than 10% of patients with laryngeal cancer are younger than 40 years of age [1].

Smoking is a leading risk factor for disease development. Other risk factors are alcohol abuse, harmful working environments—such as those with nickel fumes, asbestos, or hydrochloric acid—and chronic inflammatory diseases of the larynx [2]. Studies revealed that HPV infection could trigger laryngeal carcenogenesis, with a prevalence of HPV types 16 and 18. HPV-positive laryngeal carcinomas are most often observed in patients under 40 years of age with a good 12-month survival rate [3,4].

Laryngeal malignancy presents significant clinical challenges due to its vital impact on fundamental functions such as speech and respiration. The disease is typically classified into early stages (I and II) and advanced stages (III and IV) based on tumor size, the extent of local invasion, and regional spread, which critically influence treatment decisions and prognostic outcomes. The early stages of laryngeal squamous cell carcinoma (LSCC) offer higher chances of successful management through minimally invasive and organ-preserving therapies. In contrast, the advanced stages of LSCC often require more aggressive multimodal treatment approaches due to the presence of extensive local invasion, regional lymph node metastasis, or distant spread [5].

Despite significant advances in high-tech genetic testing related to oncological monitoring and the treatment of cancers [6]—particularly in HPV-negative LSCC—there remains a lack of specific biomarkers for cancer screening, monitoring, or therapeutic strategies.

Research has shown that the molecular pathology of HPV-negative LSCC is highly heterogeneous [7], reflecting considerable diversity in cellular and genetic characteristics both within and between individual tumors. This heterogeneity presents a major challenge in developing effective oncological therapies, as it can lead to variable tumor responses and contribute to treatment resistance [8].

MicroRNAs (miRNAs), small non-coding RNAs, have garnered significant interest as potential biomarkers. They play crucial regulatory roles in cells, and their aberrant expression has been studied extensively in solid tumors. miRNAs influence and facilitate cancer transformation and progression, showing promise for implementation in routine clinical practice [9].

Our previous studies focused on deregulated miRNAs in tissue samples from patients with HPV-negative advanced primary LSCC. The results identified two miRNAs—miR-31-3p and miR-196a-5p—that were significantly elevated in LSCC tumors compared to adjacent normal laryngeal tissue [10]. MiR-31-3p was revealed as an oncogenic factor which contributes to metastatic potential and predicts overall survival in patients treated with anti-EGFR therapies. Studies reported significantly good cetuximab responses and good overall survival in patients with metastatic colorectal cancer (mCRC) and oral squamous cell carcinoma (OSCC) [11,12]. miR-196a-5p has been linked to premalignant conditions, plays a role in the transition to malignancy in several cancer types, and it is suggested as biomarker for disease progression [13,14]. Furthermore, both miR-31-3p and miR-196a-5p demonstrated potential as non-invasive circulating biomarkers in the cohort of patients with HPV-negative advanced LSCC [15]. While these miRNAs are known to play roles in the tumorigenesis of laryngeal cancer [16], their potential as screening biomarkers in liquid biopsy in the early stages of laryngeal cancer remains limited and underexplored. Due to the significant potential of these biomarkers, they need a further evaluation.

The aim of the current study is to investigate the expression levels of miR-31-3p and miR-196a-5p in plasma samples from patients with early and advanced stages of HPV-negative primary LSCC. Additionally, this study aims to analyze their associations with clinicopathological features, co-expression, and screening value. The plasma samples from patients with advanced LSCC were collected as an independent group separate from the previous studies conducted by our team. The findings could elucidate their potential as non-invasive markers for the early detection and monitoring of laryngeal cancer.

## 2. Materials and Methods

### 2.1. Patient and Control Groups

A total of 50 patients were enrolled in this study, including 24 diagnosed with primary advanced-stage LSCC and 26 with primary early-stage LSCC. The eligibility criteria included histologically confirmed LSCC and age ≥18 years. The exclusion criteria comprised age <18 years, the presence of non-LSCC malignancies, transplant recipients, individuals with chronic infections or autoimmune diseases, and pregnant or lactating women.

For the purpose of this study, a control group of 20 healthy volunteers was also recruited. The inclusion criteria for the controls included age ≥18 years, as well as the absence of oncological conditions, chronic infections, autoimmune diseases, transplant history, pregnancy, or lactation.

The patients and controls were recruited at the Ear, Nose, and Throat Department of the University Hospital “Tsaritsa Yoanna”-ISUL, Sofia, Bulgaria, during the period from 2018 to 2024. Written informed consent was obtained from each participant and is stored at the repository of the Molecular Medicine Center and the Department of ENT, at the Medical University of Sofia. This study was approved by the Ethics Committee of the Medical University of Sofia (approval № 04-03/06.03.2018, № 56-01/12.01.2024, № 02-12/03.06.2025). None of the patients or controls had received surgery, chemotherapy, or radiotherapy prior to blood sample collection.

### 2.2. Plasma Samples

Blood samples (5–7 mL) were collected in EDTA tubes (BD Vacutainer^®^ Purple EDTA Blood Collection Tube, Aalst, Belgium) from each LSCC patient and healthy volunteer. To separate plasma from whole blood and isolate circulating miRNAs, a two-step centrifugation process was performed. This procedure aims to minimize interference from cellular debris and platelet-associated miRNAs, which can affect the accuracy of miRNA analysis.

After blood was drawn, an initial low-speed centrifugation was carried out at 1500× *g* for 10 min at room temperature (25 °C) to remove blood cells. The plasma supernatant (2 mL) was then carefully collected and transferred into a new DNase- and RNase-free tube (Thermo Fisher Scientific, MA, USA). Subsequently, high-speed centrifugation at 10,000× *g* for 15 min at 4 °C was performed to further remove cellular debris, ensuring a clear plasma sample suitable for highly sensitive miRNA analysis.

The final plasma supernatant was promptly frozen in liquid nitrogen (−196 °C) and transported to the Molecular Medicine Center Biobank in Sofia, Bulgaria, within 1–2 days. The samples were then stored at −80 °C. The Molecular Medicine Center is part of BBMRI-ERIC (European Research Infrastructure for Biobanking and Biomolecular Resources) and adheres to Good Laboratory Practices (GLP) to ensure the quality and reliability of all procedures and laboratory tests conducted.

### 2.3. Histopathological and Clinical Characteristics

During surgical treatment of the 50 patients with LSCC, formalin-fixed, paraffin-embedded (FFPE) tissue was utilized to determine the histopathological characteristics of the tumors. Fixation in neutral, buffered formalin for an appropriate time at room temperature was used. Staining with hematoxylin and eosin (HE) was performed for analysis of histopathological diagnosis. The results reported the type and origin of the tissue samples, as well as a determination of cancer stage and cancer grade. Lymph node metastasis and cell differentiation were examined histopathologically. A monoclonal-specific antibody directed against p16ink4a (CDKN2A) protein (RM409, Sigma-Aldrich, St. Louis, MO, USA) was used to analyze the presence of HPV infection. Along with histopathological characteristics, details about sex, age, tobacco smoking, alcohol use, familial history, and work exposure were taken.

### 2.4. RNA Extraction

All plasma supernatant samples included in this study were subjected to total RNA (tRNA) extraction using the miRNeasy Serum/Plasma Advanced Kit (Qiagen, Hilden, Germany), following the manufacturer’s protocol. A total of 200 µL of plasma supernatant was used for tRNA extraction. The quality and concentration of the RNA were assessed with a NanoDrop 2000 spectrophotometer (Thermo Fisher Scientific, MA, USA). All samples demonstrated high RNA purity, indicated by a favorable 260/280 ratio.

### 2.5. Reverse Transcription and Real-Time Expression Analysis

The expression profiles of circulating miR-31-3p and miR-196a-5p were analyzed in all plasma samples (from LSCC patients and controls) using quantitative real-time PCR (qPCR) with TaqMan Advanced miRNA Assays. The assays included the following: has-miR-31-3p (Accession: MI0000089; Sequence: UGCUAUGCCAACAUAUUGCCAU) and has-miR-196a-5p (Accession: MI0000238; Sequence: UAGGUAGUUUCAUGUUGUUGGG) (Thermo Fisher Scientific, MA, USA).

For each sample, 100 ng of total RNA was reverse transcribed using the TaqMan Advanced miRNA cDNA Synthesis Kit (Thermo Fisher Scientific, MA, USA), following the manufacturer’s instructions. The cDNA synthesis was followed by qPCR performed on a 7900HT Real-Time PCR platform (Thermo Fisher Scientific, MA, USA), allowing sensitive detection and analysis. Fluorescence signals were monitored over 40 amplification cycles.

The cycle threshold (Ct) values, representing the cycle at which fluorescence exceeds the threshold, were normalized to RNU6B (Assay name: RNU6B; NCBI Accession: NR_002752; Control sequence: CGCAAGGATGACACGCAAATTCGTGAAGCGTTCCATATTTTT). The relative expression (relative quantification, RQ) of each target miRNA was calculated using the 2^−ΔΔCt^ method [17]. All experiments were performed in triplicate, with at least three negative controls and one positive control included in each run.

Expression levels were interpreted as follows: RQ ≥ 2 indicated overexpression, RQ < 0.5 indicated underexpression, and an RQ between 0.5 and 1.99 indicated no significant change in expression.

### 2.6. Statistical Analysis

Statistical analyses were conducted using SPSS v27 (International Business Machines Corporation, New York, NY, USA). The Shapiro–Wilk test assessed the normality of data distribution. Descriptive statistics summarized the main features of the miRNA expression data, including measures of central tendency (mean and median), variability (standard deviation), range, and frequency distributions. Non-parametric tests—Mann–Whitney and Kruskal–Wallis—were employed to compare groups. The Receiver Operating Characteristic (ROC) analysis evaluated the diagnostic performance of the miRNAs. Spearman’s correlation assessed the relationships between variables. A *p*-value less than 0.05 was considered statistically significant. Visualizations included boxplots and scatterplots with fitted regression lines to illustrate correlations between variables.

## 3. Results

### 3.1. Clinical and Pathological Features of the Primary LSCC Patient and Healthy Control Groups

In this study, 50 patients diagnosed with primary laryngeal squamous cell carcinoma were included. Twenty-six of the patients were diagnosed with early-stage LSCC, and twenty-four with advanced LSCC. The majority were males (n = 41), and the distribution between the age groups (≤60 and >60) was slightly equal. A positive oncological familial history was reported in eight (16%) cases. All patients used tobacco and consumed alcoholic beverages. Work exposure was indicated in 21 (42%) of the patients. The control group was represented by 4 females and 16 males. The mean age of the controls was 46 years old (range 32–71). Seventeen of them reported tobacco and alcohol use, whereas five worked in a harmful environment and three of the twenty controls reported positive family oncological history.

Histopathological analysis revealed the following distribution of T stages: early T1 stage—10 patients; early T2 stage—16 patients; advanced T3 stage—13 patients; and advanced T4 stage—11 patients. The majority of cases were lymph node negative (N0)—36 patients. Cancer differentiation stages G1 and G2 were most common, with a total of 44 patients. All the included patients were evaluated with p16 immunostaining, which was negative, indicating HPV-negative tumors.

All clinical and pathological data are summarized in Table 1.

### 3.2. Relative Expression of Circulating miR-31-3p and miR-196a-5p in Primary HPV-Negative Plasma LSCC Samples

The RNA expression levels of free-circulating miR-31-3p and miR-196a-5p were assessed in all the samples included in this study, totaling 50 plasma samples from LSCC-diagnosed patients and 20 healthy volunteers. The miRNA expression levels in the LSCC patients were compared to the average miRNA levels in the control cohort. The normal distribution of the data for both miRNAs was evaluated using the Shapiro–Wilk test, which rejected the null hypothesis (H_0_) for normally distributed data (*p* < 0.001). Therefore, it was assumed that the data do not follow a normal distribution. Consequently, non-parametric statistical tests were used as appropriate.

In all LSCC patients, the relative expression (relative quantification, RQ) of miR-31-3p and miR-196a-5p was significantly upregulated compared to the control group (*p* < 0.001). The mean ± SD RQ of miR-31-3p in the control group was 1.07 ± SD 0.71 (median = 0.79), whereas in the early LSCC group, the mean ± SD RQ was 3.04 ± SD 2.34 (median = 2.52), and in the advanced LSCC group, the mean ± SD RQ was 8.57 ± SD 7.05 (median = 6.73) (Figure 1A). The mean ± SD expression level of miR-196a-5p in the control group was 1.01 ± SD 0.83 (median = 0.71). In the early LSCC group, the mean ± SD miR-196a-5p RQ was 4.89 ± SD 7.32 (median = 1.49), and in the advanced LSCC group, the mean ± SD was 2.99 ± SD 1.42 (median = 2.89) (Figure 1B). The Kruskal–Wallis statistical test was used to compare the three groups, including the control, early stage LSCC, and advanced sage LSCC groups.

### 3.3. Association Between miR-31-3p/miR-196a-5p Expression and Clinicopathological Features

The Mann–Whitney and Kruskal–Wallis statistical analyses revealed that miR-31-3p expression levels were significantly associated with T stage (*p* < 0.001) and N stage (*p* = 0.009), whereas miR-196a-5p was not related to the patient features included in the analysis (Table 2).

Figure 2 shows box-whisker plots representing the distribution of miR-31-3p across T stages (Figure 2A) and N stages (Figure 2B). The Mann–Whitney U and Kruskal–Wallis non-parametric tests were used for the association statistics. The median RQ values of miR-31-3p across T stages were as follows: T1—2.91, T2—2.00, T3—7.66, and T4—9.36. In terms of nodal status, the RQ value in the N0 (nodal-negative) samples was 2.82, while in the N1–3 (nodal-positive) samples, it was 8.90.

### 3.4. Evaluation of the Diagnostic Potential of Circulating miR-31-3p and miR-196a-5p

We employed a receiver operating characteristic (ROC) curve analysis to assess the diagnostic accuracy and discriminatory capacity of miR-31-3p and miR-196a-5p, alone and in combination, in distinguishing early and advanced laryngeal squamous cell carcinoma (LSCC) patients from healthy controls (Figure 3). The optimal cutoff values for relative quantification (RQ) were determined based on the highest combined sensitivity and specificity for each miRNA.

Our results showed that miR-31-3p alone demonstrated superior discriminatory power compared to miR-196a-5p and the combination of miR-31-3p and miR-196a-5p in an miRNA expression panel. Specifically, miR-31-3p significantly differentiated early-stage LSCC from the healthy controls (*p* = 0.001), with a sensitivity of 64% and a specificity of 85% at an RQ cutoff of 1.98. The area under the ROC curve (AUC) was 0.798 (95% confidence interval [CI]: 0.673–0.924).

Similarly, miR-196a-5p effectively distinguished early-stage LSCC from the controls (*p* = 0.035), with an AUC of 0.682 (95% CI: 0.527–0.837) at an RQ cutoff of 1.95, yielding a sensitivity of 56% and a specificity of 81%.

Regarding the differentiation between the early and advanced stages of LSCC, miR-31-3p showed significant discrimination (*p* < 0.001), with an AUC of 0.850 (95% CI: 0.743–0.956) at an RQ cutoff of 2.03, achieving a sensitivity of 95.5% and a specificity of 64%. In contrast, miR-196a-5p’s ability to distinguish early- from advanced-stage LSCC was not statistically significant (*p* = 0.255), with an AUC of 0.597 (95% CI: 0.428–0.766).

Additionally, the combination of miR-31-3p and miR-196a-5p in a miRNA expression panel could evaluate and screen significantly early-stage LSCC (*p* < 0.001) at a cut-off RQ = 1.89 with sensitivity of 88.5%, a specificity of 53%, and an AUC = 0.811 (95% CI: 0.690–0.933). Whereas the RQ panel of both miRNAs distinguish advanced-stage LSCC from early-stage LSCC with the highest sensitivity of 100% and the lowest specificity of 32% at higher RQ levels equal to 3.23. The calculated AUC was 0.725 (95% CI: 0.604–0.885, *p* = 0.004).

These findings suggest that miR-31-3p has greater potential as a biomarker for the early detection and staging of LSCC compared to miR-196a-5p. Intriguingly, the combination of both miRNAs in a panel also could improve the screening and diagnostic potential of early-stage LSCC.

### 3.5. Co-Expression of miR-31-3p and miR-196a-5p in LSCC

Spearman’s rho correlation coefficient was used to assess their co-expression and whether both circulating miRNA variables tend to change in the same direction. The statistical results revealed a significant, moderate-to-strong positive correlation with a correlation coefficient of (r_s_) = 0.508 (*p* = 0.001). Figure 4 presents the scatter plot distribution with a fitted line illustrating the relationship between the miR-31-3p and miR-196a-5p RQ expression values. The results indicate a lack of linearity between the two miRNAs, as demonstrated by the Pearson correlation coefficient (r = 0.053, R^2^ = 0.003, *p* = 0.685).

## 4. Discussion

A liquid biopsy is a minimally invasive diagnostic procedure that involves analyzing biological fluids to detect and monitor disease-related biomarkers. It provides valuable information for diagnosis, prognosis, and treatment monitoring, particularly in cancer management [18]. miRNAs are signature molecules that can be used in liquid biopsy analysis.

The aim of the current study was to explore the aberrant expression levels of two miRNAs, miR-31-3p and miR-196a-5p, which have been previously identified as potential biomarkers in laryngeal cancer, specifically in HPV-negative early and advanced laryngeal squamous cell carcinoma (LSCC) [11].

Our knowledge about circulating, non-invasive miR-31-3p and miR-196a-5p expression levels is very limited. This study is the first to examine the potential of both miRNAs in a very homogeneous group of head and neck cancer patients, particularly those with HPV-negative LSCC in early and advanced stages.

The results revealed that both miRNAs are detected within normal quantitative ranges in the control group. However, they were found to be elevated in both the early and advanced HPV-negative LSCC groups (*p* < 0.001). miR-31-3p showed a significant graded increase across T stages, and its expression in lymph node-positive samples (N1-3) was significantly higher compared to lymph node-negative samples (N0). Conversely, miR-196a-5p was elevated in early-stage LSCC but did not show significant differences across various pathological features. Although miR-196a-5p was determined to be nonsignificant between tumor stages, its role still persists. The published study revealed its overexpression in premalignant lesions and cancer cell differentiation [13]. Likely, the role of miR-196a-5p is significant and occurs in the very early stages of cancerogenesis, on micromalignancy levels, when screening and diagnostic tools are limited.

miR-31-3p has been extensively studied across different cancers, functioning as an oncogene (oncomiR). Its oncogenic role involves promoting tumor proliferation and invasion by suppressing targets such as LATS2, which facilitates cell proliferation [19], and FIH, which aids in hypoxia-driven tumor progression [11]. Additionally, miR-31-3p downregulates RASA1 (p120 Ras GTPase-activating protein), leading to increased activity of the Ras pathway [20].

In head and neck cancers, miR-31 is often overexpressed, especially in HPV-negative LSCC, where its expression correlates with aggressive tumor behavior and poor prognosis [21,22]. For instance, Wang LL et al. (2018) investigated miR-31-3p expression via RT-qPCR in tissue and serum samples from head and neck cancer patients. Their findings support the current study, showing significant associations between miR-31-3p levels and tumor and nodal stages [21]. Elevated miR-31-3p levels have also been linked to resistance to anti-EGFR therapies in colorectal cancer, potentially through the regulation of downstream signaling pathways such as RAS/RAF/MEK/ERK, which can bypass EGFR inhibition [22]. Since anti-EGFR targeted therapies are a part of LSCC treatment protocols, measuring miR-31-3p levels could help identify the patients unlikely to benefit from such therapies and aid in optimizing treatment strategies.

Several studies have reported increased levels of miR-196a-5p in head and neck squamous cell carcinoma (HNSCC). Generally, miR-196a-5p is involved in promoting oncogenic processes, including tumor growth, invasion, and metastasis, especially in HPV-negative laryngeal cancer. Its target pathways involve differentiation and apoptosis. The expression levels of miR-196a-5p may correlate with tumor stage, grade, or patient prognosis [23,24].

Elevated levels of miR-196a-5p are associated with chemotherapy resistance. This study suggests that suppressing miR-196a-5p using antisense oligonucleotides (antagomirs) or locked nucleic acid (LNA) inhibitors may enhance tumor sensitivity to chemotherapy [25]. Notably, the primary gene targets of miR-196a-5p include members of the HOX gene family (HOXB8, HOXC8, HOXD10), which are transcription factors involved in cell patterning and development [26]. Other targets include Annexin A1 (ANXA1), involved in anti-inflammatory responses and the regulation of cell growth; Sonic hedgehog (SHH), a key signaling molecule in developmental pathways; and ZEB2, a transcription factor implicated in epithelial–mesenchymal transition (EMT) [27,28]. miR-196a-5p regulates various genes, particularly members of the HOX family, as well as genes involved in cell growth, differentiation, and apoptosis.

miR-196a-5p is deregulated in the early stages of tongue squamous cell carcinoma (TSCC) and has been identified as a marker for delayed lymph node metastasis development [29]. Our study also supports the observation of elevated miR-196a-5p levels in the early stages of squamous carcinoma, although we did not find a statistically significant correlation between miRNA expression and pathological features. In our early laryngeal squamous cell carcinoma (LSCC) cohort, all patients were diagnosed without lymph node metastasis. We believe that the deregulation of miR-196a-5p may begin very early in the process of carcinogenesis.

Bao et al. (2018) reported that miR-195a-5p could serve as a supportive biomarker alongside other markers used in routine clinical monitoring [30].

For the first time, our data showed that miR-31-3p and miR-196a-5p co-expressed and significantly correlated positively (*p* = 0.001), which might indicate coregulation, as part of the same cellular pathway or similar biological processes. Some studies have explored the expression patterns of multiple miRNAs simultaneously [31], but specific correlation analyses between miR-31-3p and miR-196a-5p are not well established.

Certain limitations should be recognized to accurately interpret our findings, notably the small sample size and the limited number of participants in the control group. Although recent progress has been made in liquid biopsy techniques, several technical challenges persist. These include issues related to proper sample handling, the absence of standardized protocols for isolating and normalizing miRNAs, and regulatory obstacles, all of which hinder its widespread adoption in clinical medicine. In addition, the lack of in vivo analysis restricts a deeper understanding of miR-31-3p’s and miR-196a-5p’s cell regulatory roles.

Our next goal is to implement the measurement of miR-31-3p and miR-196a-5p expression levels via RT-qPCR as non-invasive screening markers, to be used alongside clinical tests or methods for disease assessments and diagnosis. The implications and improvement of such biomarkers would contribute to the optimization of laryngeal cancer treatment strategies, patients’ quality of life, and survival rates

## 5. Conclusions

In conclusion, the current study revealed that miR-31-3p could serve as a liquid biopsy biomarker for screening of the early stages of LSCC. The published studies related to circulating miRNAs in laryngeal cancer are limited. The current research is focusing on plasma miRNAs as potential biomarkers, particularly in HPV-negative primary early and advanced LSCC stages. The main strength of this study is that the results confirmed the data from our previous projects on miRNA expression analysis and highlighted miR-31-3p as a significant variable with promising use in routine clinical practice for early LSCC assessment. The results about miR-196a-5p as a standalone also revealed its overexpression in early laryngeal cancer stages, and this miRNA also has the capacity to be a monitoring marker. But still, this study meets some limitations. Our next goal is to enlarge the target LSCC patient and control groups and analyze the encouraging results in bigger cohort samples in order to validate the data. Moreover, as a second goal is to set a protocol of implementation of miR-31-3p and/or miR-196a-5p as surrogate markers in clinical practice for early LSCC recurrence.

## Figures and Tables

**Figure 1 diseases-13-00279-f001:**
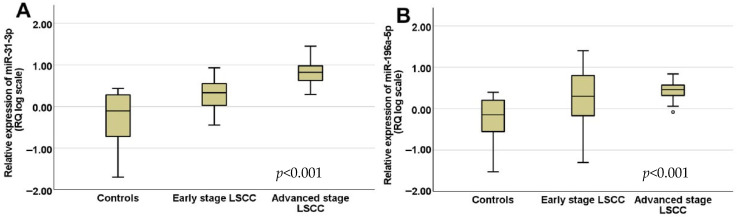
The whisker-boxplot of circulating (**A**) miR-31-3p and (**B**) miR-196a-5p across the controls, and the early, and advanced LSCC patient groups. The expression levels of (**A**) miR-31-3p and (**B**) miR-196a-5p were significantly deregulated between the three groups of plasma samples. The level of significance for both miRNAs was *p* < 0.001 (Kruskal–Wallis statistical test).

**Figure 2 diseases-13-00279-f002:**
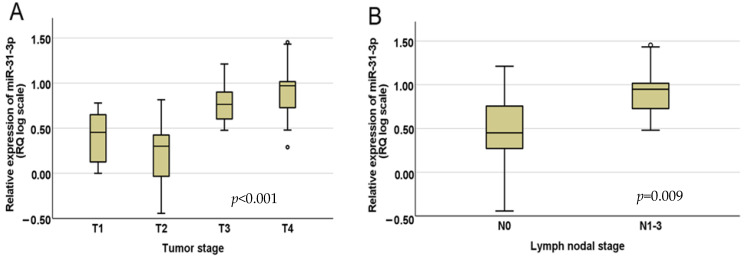
Distribution of miR-31-3p between (**A**) tumor stages and (**B**) nodal negative and positive stages.

**Figure 3 diseases-13-00279-f003:**
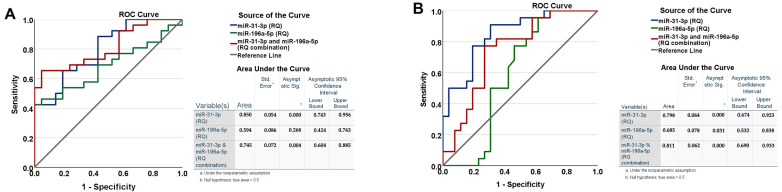
The screening potential of miR-31-3p and miR-196a-5p alone and in combination between (**A**) early-stage LSCC patients and healthy controls and (**B**) early-stage LSCC patients and advanced-stage LSCC patients. RQ, Relative Quantification; ROC, receiver operating characteristic; AUC; Area Under the ROC Curve.

**Figure 4 diseases-13-00279-f004:**
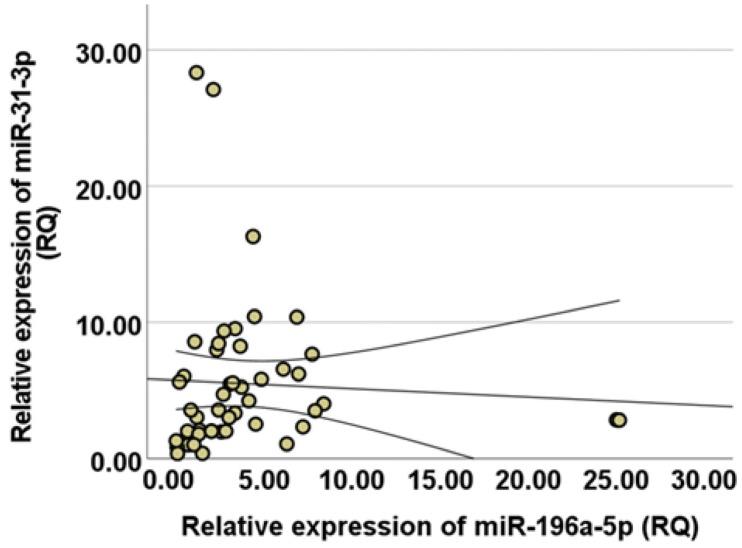
Spearman correlation between circulating miR-31-3p and miR-196a-5p showed positive co-expression with the correlation coefficient rs = 0.508 (*p* = 0.001). The fitted line was added to model the relationship between both variables.

**Table 1 diseases-13-00279-t001:** Features of 50 patients diagnosed with primary laryngeal squamous cell carcinoma and 20 healthy controls.

Features	LSCC Patients n (%)	Healthy Controls n (%)
Gender		
Female	9 (19)	4 (20)
Male	41 (81)	16 (80)
Age		
≤60	24 (48)	9 (45)
>60	26 (52)	11 (55)
Tobacco		
Yes	50 (100)	17 (85)
No	0 (0)	3 (15)
Alcohol		
Yes	50 (100)	17 (85)
No	0 (0)	3 (15)
Work exposure		
Yes	21 (42)	5 (25)
No	29 (58)	11 (75)
Family history		
Yes	8 (16)	3 (15)
No	42 (84)	17 (85)
Tumor stage		
T1	10 (21)	
T2	16 (34)	
T3	13 (22.5)	
T4	11(22.5)	
Nodal stage		
N0	36 (75)	
N1–3	14 (25)	
G stage		
G1	20 (40)	
G2	24 (48)	
G3	6 (12)	
HPV (p16 staining)		
Positive	0 (0)	
Negative	50 (100)	

**Table 2 diseases-13-00279-t002:** Association between miR-31-3p and miR-196a-5p and the clinicopathological features of a cohort of 50 patients diagnosed with primary laryngeal squamous cell carcinoma.

Clinicopathological Features	LSCC Patients n (%)	miR-31-3p RQ (Mean ± SD)	miR-196a-5p RQ (Mean ± SD)
Gender			
Female	9 (19)	6.45 (4.75)	4.02 (5.41)
Male	41 (81)	5.24 (5.98) *p* = 0.272	2.42 (2.20) *p* = 0.599
Age			
≤60	24 (48)	4.49 (4.08)	4.50 (6.71)
>60	26 (52)	6.44 (6.98) *p* = 0.187	2.94 (2.17) *p* = 0.697
Tumor stage			
T1	10 (21)	3.10 (1.94)	2.63 (3.11)
T2	16 (34)	2.12 (1.61)	4.76 (8.12)
T3	13 (22.5)	9.57 (7.69)	3.67 (2.17)
T4	11(22.5)	10.30 (9.37) *p* < 0.001	3.05 (1.60) *p* = 0.223
Nodal stage			
N0	36 (75)	3.78 (3.20)	3.98 (5.68)
N1–3	14 (25)	10.53 (8.43) *p* = 0.009	2.92 (1.65) *p* = 0.860
G stage			
G1	20 (40)	4.08 (4.01)	3.89 (5.63)
G2	24 (48)	5.80 (5.68)	3.86 (5.16)
G3	6 (12)	8.55 (9.52) *p* = 0.293	2.64 (1.21) *p* = 0.910

n, number; RQ—relative quantification; SD—standard deviation.

## Data Availability

The data presented in this study are available on request from the corresponding author.

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
