# Peer review of "Integration of Circulating miR-31-3p and miR-196a-5p as Liquid Biopsy Markers in HPV-Negative Primary Laryngeal Squamous Cell Carcinoma"

_diseases, 2025, doi:10.3390/diseases13090279_

Round 1

Reviewer 1 Report

Comments and Suggestions for Authors

In their research, the authors focused on the use of two circulating blood miRNAs as markers for HPV-negative forms of laryngeal cancer.
Several comments are addressed to the authors:

  1. The authors justify their choice of miRNAs by referring to their previous publication. Nevertheless, they should also explain and justify this choice more clearly in the introduction of the current paper, particularly why these miRNAs are worth further evaluation.

  2. The main weakness of the study is the small size of both the study and control groups, which may limit the reliability of the conclusions.

  3. The authors should present a detailed characterization of the patients included in the study in the form of a table.

  4. The methodology should be supplemented with information such as the volume of blood used for plasma extraction, the volume used for RNA isolation, etc.

  5. Figures 1 and 2 – applying a logarithmic scale to the y-axis may improve the clarity of the graphs. Additionally, please indicate the p-values and mention them in the text.

  6. Table 2 – the T, G, and N features should correspond to the rows in the subsequent part of the table.

  7. Figure 3 – a combined analysis of both miRNAs would be useful, along with an assessment of the diagnostic value of such a panel.

  8. All study limitations should be thoroughly described.

  9. The conclusions are too brief; they should highlight the strengths of the study, identify areas for improvement, point out key observations, and suggest directions for future research.

  10. The manuscript contains numerous language errors – the text should be revised by a qualified language professional.

Author Response

  1. The authors justify their choice of miRNAs by referring to their previous publication. Nevertheless, they should also explain and justify this choice more clearly in the introduction of the current paper, particularly why these miRNAs are worth further evaluation.

Response 1: Thank you for your comment. I agree. In the introduction was included additional details why these miRNAs are in interest and why do they need a further evaluation. The changes could be found from line 80 to 86.

  1. The main weakness of the study is the small size of both the study and control groups, which may limit the reliability of the conclusions.

Response 2: Thank you for pointing this out. Limitations of the study were added in the discussion manuscript part.

  1. The authors should present a detailed characterization of the patients included in the study in the form of a table.

Response 3: Thank you for your comment. A new and detailed Table with patients and controls’ features was added for more clear understanding (Table 1).

  1. The methodology should be supplemented with information such as the volume of blood used for plasma extraction, the volume used for RNA isolation, etc.

Response 4: Thank you for your comment. The sample volumes were added in the manuscript.

  1. Figures 1 and 2 – applying a logarithmic scale to the y-axis may improve the clarity of the graphs. Additionally, please indicate the p-values and mention them in the text.

Response 5: Thank you for your comment. Agree. The RQ values were logarithmic scaled and the p values were included in the Figure 1 and 2. The significance and p-values were included in the text – at lines 233/234 and 248.

  1. Table 2 – the T, G, and N features should correspond to the rows in the subsequent part of the table.

Response 6: Thank you for your comment. The table was corrected.

  1. Figure 3 – a combined analysis of both miRNAs would be useful, along with an assessment of the diagnostic value of such a panel.

Response 7:  A combined analysis of both miRNAs is done and included in the results, paragraph 3.4.

  1. All study limitations should be thoroughly described.

Response 8:  Thank you for your comment. Study’s limitations were included in the discussion.

  1. The conclusions are too brief; they should highlight the strengths of the study, identify areas for improvement, point out key observations, and suggest directions for future research.

Response 9:  Thank you. The conclusion is improved.

  1. The manuscript contains numerous language errors – the text should be revised by a qualified language professional.

Response 10:  Thank you for your comment. The manuscript would be revised by a journal’s professional for correction.

Reviewer 2 Report

Comments and Suggestions for Authors

This study analyzed the levels of circulating miR-31-3p and miR-196a-5p in plasma samples of patients with HPV-negative primary laryngeal squamous cell carcinoma. They found miR-31-3p is up-regulated in early and advanced HPV-negative laryngeal cancer patients and correlated with tumor stage and nodal stage. They claimed that circulating miR-31-3p and miR-196a-5p as liquid biopsy markers in HPV-Negative primary laryngeal squamous cell carcinoma. It may provide useful biomarker for LSCC screening. However, there are many problems.

  1. Figure 1. The number of cases in each group should be added in legend. The results of statistical analysis should be added in the figure. Please provide detail statistical method about this figure.
  2. Figure 1. In figure legend, author said “The expression levels of (A) miR-31-3p and (B) miR-196a-5p was significantly deregulated between the three groups of plasma samples”. It is confusing. How was relative quantification (RQ) calculated? This should be clearly explained in method. The relative expression of each target miRNA was calculated using the 2^-ΔΔCt method. Authors should explain how they calculate the relative expression level in detail. Please provide the original data of Figure 1.
  3. Table 1 showed miR-196-5p has no significant association with LSCC’s tumor stage, nodal stage, and G stage. Figure 3 and Line 250 showed that miR-196a-5p's ability to distinguish early from advanced LSCC was not statistically significant (p=0.255). Therefore, these results cannot support the conclusion of miR-196a-5p has potential as LSCC screening markers.
  4. Figure 4 legend: “Spearman correlation between circulating miR-31-3p and miR-196a-5p showed positive co-expression with correlation coefficient rs=0.508 (p=0.001). The fitted line was added to models the relationship between both variables.” However, Figure 4 didn’t show good correlation between miR-31-3p and miR-196-5p. Please provide the original data of figure 4.
  5. Line 284. “The results revealed that both miRNAs are detected within normal quantitative ranges in the control group.” What are the normal quantitative ranges of these miRNAs?
  6. Line 286. “They demonstrated oncogenic roles, with upregulation even in very early stages (T1) of LSCC.” In this manuscript, authors have no data to demonstrate these miRNAs have oncogenic roles.
  7. The title is “Integrating of Circulating miR-31-3p and miR-196a-5p as Liquid Biopsy Markers in HPV-Negative Primary Laryngeal Squamous Cell Carcinoma”. However, no data support integrating analysis by using both miRNAs simultaneously.

The writing is very poor. There are many mistakes. For example:

Line 197. “The miRNA expression levels in LSCC patients were matched to control individuals based on sex and age.” It is wrong. How could miRNA expression level in a group match another group?

“The mean RQ of miR-31-3p in the control group was 1.07 ± SD 0.71 (median = 0.79),” There are many similar statements in the whole manuscript. It is wrong. How could mean be expressed as mean ± SD?

Line 280, “Our knowledge about circulating, non-invasive miR-31-3p and miR-196a-5p expression levels is very limited”. What is “non-invasive miR-31-3p and miR-196a-5p expression levels”? Is there any invasive miR-31-3p and miR-196a-5p expression levels?

Line 274, “miRNAs are molecular fragments that can be used in liquid biopsy 274 analysis.” What is molecular fragments? Why are miRNAs molecular fragments?

Line 276-278, “The aim of the current study was to explore the aberrant expression levels of two miRNAs, miR-31-3p and miR-196a-5p, which have been previously identified as potential biomarkers in laryngeal cancer,” Please add a reference related to “previously”.

Table 1, first line, “LSCC pstients” should be “LSCC patients”.

Check the whole manuscript carefully and revise all mistake. Highlight all revisions.

Author Response

  1. Figure 1. The number of cases in each group should be added in legend. The results of statistical analysis should be added in the figure. Please provide detail statistical method about this figure.

Response 1: Thank you for your comment: Figure 1 is improved. SPSS software was used to be created the boxplots. You could see that X-axis represents sample types: controls, early LSCC and advanced LSCC stages, whereas Y-axis miR-31-3p or miR-196a-5p levels.

  1. Figure 1. In figure legend, author said “The expression levels of (A) miR-31-3p and (B) miR-196a-5p was significantly deregulated between the three groups of plasma samples”. It is confusing. How was relative quantification (RQ) calculated? This should be clearly explained in method. The relative expression of each target miRNA was calculated using the 2^-ΔΔCt method. Authors should explain how they calculate the relative expression level in detail. Please provide the original data of Figure 1.

Response 2: Thank you for your comment. The RQ values of each miRNA are calculated by 2^-ΔΔCt method, which is published and a citation is included in the manuscript. The method is simple and very precise.

Please, see the ref.:

Livak KJ, Schmittgen TD. Analysis of relative gene expression data using real-time quantitative PCR and the 2(-Delta Delta C(T)) Method. Methods. 2001 Dec;25(4):402-8. doi: 10.1006/meth.2001.1262. PMID: 11846609;

Please, provide me an email and I will send to you all data and calculations. I will send the files also to the Journal’s editor in Cc.

  1. Table 1 showed miR-196-5p has no significant association with LSCC’s tumor stage, nodal stage, and G stage. Figure 3 and Line 250 showed that miR-196a-5p's ability to distinguish early from advanced LSCC was not statistically significant (p=0.255). Therefore, these results cannot support the conclusion of miR-196a-5p has potential as LSCC screening markers.

Response 3: Agree. The manuscript and conclusion are improved.

  1. Figure 4 legend: “Spearman correlation between circulating miR-31-3p and miR-196a-5p showed positive co-expression with correlation coefficient rs=0.508 (p=0.001). The fitted line was added to models the relationship between both variables.” However, Figure 4 didn’t show good correlation between miR-31-3p and miR-196-5p. Please provide the original data of figure 4.

Response 4: These are the original data. I am sorry, but I do not understand your point. According to our long-term research projects, miR-31-3p could be very promising marker. Its expression was few times validated in separate patients’ group, together with previous pilot project of plasma samples from advanced stage LSCC patients. This project is a continuation of our pilot plasma project, and miR-31-3p again showed significant results not only in new collected advanced laryngeal cancer patients, but in early-stage laryngeal cancers. About miR-196a-5p, our experience showed that this miRNA is definitely expressed in early stages of laryngeal cancer in comparison to the controls, but there is not significant difference between early and advanced laryngeal stages. Another research team published that miR-196a-5p expression is overexpressed in premalignant lesions and could drive carcenogenesis processes. Our hypothesis about miR-196a-5p is that, the miRNAs have a key role in microcancerogenesis stages, and definitely worth its exploration. Mir-196a-5p may have the potential of very early malignant marker, but additional project must be carried out.

Please, provide me an email and I will send to you all data and calculations. I will send the files also to the Journal’s editor in CC.

  1. Line 284. “The results revealed that both miRNAs are detected within normal quantitative ranges in the control group.” What are the normal quantitative ranges of these miRNAs?

Response 5: Thank you for your comment. The normal RQ range according to our real time software is between 0.5 and 1.99. The calculations vary between real-timePCR platforms and used software.

  1. Line 286. “They demonstrated oncogenic roles, with upregulation even in very early stages (T1) of LSCC.” In this manuscript, authors have no data to demonstrate these miRNAs have oncogenic roles.

Response 6: Thank you for your comment and you pointed it out. According to American Cancer Society the definition of oncogene is “When a proto-oncogene mutates (changes) or there are too many copies of it, it can become turned on (activated) when it is not supposed to be, at which point it's now called an oncogene. When this happens, the cell can start to grow out of control, which might lead to cancer.

Oncogenes can be turned on (activated) in cells in different ways. For example:

Gene variants/mutations:……

Epigenetic changes:……

Chromosome rearrangements:…….

Gene duplication:……”

The overexpression of both miRNAs is definitely, and they participate in malignancy processes. However, why exactly they are overexpressed is not known in our included in the project patients. This could be project in the future. How miR-31-3p and miR-196a-5p become an oncogenes, even in our early stages of LSCC.

  1. The title is “Integrating of Circulating miR-31-3p and miR-196a-5p as Liquid Biopsy Markers in HPV-Negative Primary Laryngeal Squamous Cell Carcinoma”. However, no data support integrating analysis by using both miRNAs simultaneously.

Response 7: thank you for your comment. The manuscript is improved, and additional analysis is added with combination of both miRNAs. Please, check Figure 3.

The writing is very poor. There are many mistakes. For example:

Line 197. “The miRNA expression levels in LSCC patients were matched to control individuals based on sex and age.” It is wrong. How could miRNA expression level in a group match another group?

Response: Thank you for pointing this out. We aimed to compare the miRNA expression. We discussed in out team the detail. Unfortunately, a mistake was done during manuscript writing, and this information was not written proper. We compared the miRNA expression levels in the target LSCC patients to the average miRNA expression levels in the control group. The both miR-31-3p and miR-196a-5p showed expression in the RQ ranges without outliers. The sentence was corrected. Thank you.

“The mean RQ of miR-31-3p in the control group was 1.07 ± SD 0.71 (median = 0.79),” There are many similar statements in the whole manuscript. It is wrong. How could mean be expressed as mean ± SD?

Response: Thank you for your comment, but unfortunately, I do not understand the issue. The mean summarizes the data and standard deviation (SD) quantifies the spread or variability of the data points around the mean. I included the median as well, due to that nonparametric statistical tests were used. And I wanted to be as much as correct with the obtained data.

Line 280, “Our knowledge about circulating, non-invasive miR-31-3p and miR-196a-5p expression levels is very limited”. What is “non-invasive miR-31-3p and miR-196a-5p expression levels”? Is there any invasive miR-31-3p and miR-196a-5p expression levels?

Response. Thank you for your comment. Of course, invasive miR-31-3p or miR-196a-5p can be considered. For example, if miR-31-3p or miR-196a-5p is to be explored in tissue samples (such as needle biopsies or surgical specimens), this involves analysis of invasive expression. This approach could be helpful when using a miRNA marker for proper characterization of the field of cancerization.

Line 274, “miRNAs are molecular fragments that can be used in liquid biopsy 274 analysis.” What is molecular fragments? Why are miRNAs molecular fragments?

Response. Thank you for your comment. The used phrase “molecular fragments” was not used properly. In the sentence “molecular fragments” was replaced with “signature molecules”. I tried to find a synonym of biomarker.

Line 276-278, “The aim of the current study was to explore the aberrant expression levels of two miRNAs, miR-31-3p and miR-196a-5p, which have been previously identified as potential biomarkers in laryngeal cancer,” Please add a reference related to “previously”.

Response: Thank you. The reference is added.

Table 1, first line, “LSCC pstients” should be “LSCC patients”.

Response: Agree. It is corrected.

Check the whole manuscript carefully and revise all mistake. Highlight all revisions.

Response: Agree. Thank you for your comment. The manuscript would be revised by a journal’s professional for correction.

Reviewer 3 Report

Comments and Suggestions for Authors

1. Why is the focus on HPV-negative laryngeal cancer? The mechanism of miR-31-3p and miR-196a-5p involvement in carcinogenesis will differ depending on the HPV status?
2. Why is blood chosen as a biological fluid? Perhaps the content of miR-31-3p and miR-196a-5p in saliva would be more informative, given that local accumulation is possible?
3. For miR-196a-5p, the diagnostic significance is still unclear. It is definitely not worth focusing on it. However, it is quite possible that a combination of two miRNAs will provide new information. Have the authors tried to look at the level of miR-196a-5p with low and high levels of miR-31-3p?
4. There is no information on healthy volunteers, in particular the structure of the group by gender and age. It is only stated that the age is over 18 years, but in the main group there are patients over 60 years old, so statistically significant differences can be due to the age difference at least. Add the relevant information to the manuscript text.

Author Response

  1. Why is the focus on HPV-negative laryngeal cancer? The mechanism of miR-31-3p and miR-196a-5p involvement in carcinogenesis will differ depending on the HPV status?

Response 1: Thank you for your comment.

Our research team is working more than a decade in the study projects related to the improving knowledge about molecular mechanisms that underlying in laryngeal cancerogenesis, and more particularly in laryngeal squamous cell carcinoma.

It is known that HPV infection directly influences miRNA expression. Both miRNA, which are the focus of our study – miR-31-3p and miR-196a-5p are published as affected by HPV infection with high-risk virus subtypes. Their expression levels are decreased due to HPV in cervical cancer and cell lines. Whereas in majority of research projects related to other solid tumors report their elevated levels during cancerogenesis.

By itself HPV infection changes the molecular characteristics of cells, and they could become malignant. In oral cancer, HPV infections are more common than in laryngeal cancer. Studies reported elevated miR-31-3p expression in oral cancer, but the parallel expose of other environmental factors could not be neglected as tobacco smoking, especially along with alcohol use. These factors lead to main malignant transformation. Still the knowledge about miR-31-3p underlying mechanisms should be deeply investigated in order to examine the best biomarkers in different cancer subtypes, including those who are HPV-positively solely in patient who do not smoke and drink beverages.

HPV16 subtype decrease levels of miR-196a-5p in cervical cancer, but is found overexpressed in HPV-positive oral squamous cell carcinoma. In the study of Yunxia Wan et al (2017) is shown that miR-196a is found overexpressed in HPV negative and HPV positive head and neck cancers. Still, the exposure of various environmental factors should be considered: HPV infection, in combination with or without tobacco or/and alcohol use.

At the moment a project of our team is running, related to comparison of both miRNA, and not only, in different laryngeal cancer subgroups, including HPV negative and HPV-positive LSCC.

Ref:

Liu, C., Lin, J., Li, L. et al. HPV16 early gene E5 specifically reduces miRNA-196a in cervical cancer cells. Sci Rep 5, 7653 (2015). https://doi.org/10.1038/srep07653

Rodrigues P, Rizaev JA, Hjazi A, Altalbawy FMA, H M, Sharma K, Sharma SK, Mustafa YF, Jawad MA, Zwamel AH. Dual role of microRNA-31 in human cancers; focusing on cancer pathogenesis and signaling pathways. Exp Cell Res. 2024 Oct 1;442(2):114236. doi: 10.1016/j.yexcr.2024.114236. Epub 2024 Sep 6. PMID: 39245198.

Lin, X., Wu, W., Ying, Y. et al. MicroRNA-31: a pivotal oncogenic factor in oral squamous cell carcinoma. Cell Death Discov. 8, 140 (2022). https://doi.org/10.1038/s41420-022-00948-z

Wan Y, Vagenas D, Salazar C, Kenny L, Perry C, Calvopiña D, Punyadeera C. Salivary miRNA panel to detect HPV-positive and HPV-negative head and neck cancer patients. Oncotarget. 2017 Oct 10;8(59):99990-100001. doi: 10.18632/oncotarget.21725. PMID: 29245955; PMCID: PMC5725146.).

  1. Why is blood chosen as a biological fluid? Perhaps the content of miR-31-3p and miR-196a-5p in saliva would be more informative, given that local accumulation is possible?

Response 2: Thank you, that you pointed that.

Previously, we have a project related to exploration of four miRNAs only in advanced HPV-negative LSCC. In this project is included a new separately enrolled patient’s group diagnosed with advanced LSCC and novel patient group diagnosed with early LSCC. One of the goals was to validate the previous results in new samples. This is the main reason, why we choose to include and work with blood samples, as a continuation of the previous projects. Additionally, blood-based miRNAs can be useful for detecting systemic diseases. Still, some limitations persist with salivary collection. The saliva samples would give us mainly information about oral location, and sampling could be influenced by external factors, as food or beverages consummation etc. Must be pointed out, that the research studies related to exploration and comparison of non-invasive samples are limited.

For the future we would validate the promising miR-31-3p data in non-invasive sampling, as saliva.

  1. For miR-196a-5p, the diagnostic significance is still unclear. It is definitely not worth focusing on it. However, it is quite possible that a combination of two miRNAs will provide new information. Have the authors tried to look at the level of miR-196a-5p with low and high levels of miR-31-3p?

Response 3: Thank you for your comment. The manuscript is improved and additional analysis of both miRNAs in combination as diagnostic marker is added. The Figure 3 is improved. However, I am not agreeing with “It is definitely not worth focusing on it.” It is published, that miR196a-5p is overexpressed in premalignant lesions and in very beginning processes of malignancy. miR-196a-5p is know miRNAs, that is strong regulator of differentiation processes and I believe, that its deregulation started during microcancerogenesis stages. Definitely, the continuation of the miR-196a-5p research in the field worth it in order to explore its specific role.

  1. There is no information on healthy volunteers, in particular the structure of the group by gender and age. It is only stated that the age is over 18 years, but in the main group there are patients over 60 years old, so statistically significant differences can be due to the age difference at least. Add the relevant information to the manuscript text.

Response 4: Agree. Details about controls were included in the manuscript and a new Table (Table 1) was included with all

features of the patients and controls.

Reviewer 4 Report

Comments and Suggestions for Authors

Dear Authors,

The manuscript explores the diagnostic potential of two circulating microRNAs miR-31-3p and miR-196a-5p as liquid biopsy markers in HPV-negative laryngeal squamous cell carcinoma (LSCC). The study is timely and relevant, given the ongoing search for non-invasive biomarkers in head and neck cancers. The methodology is sound, and the results provide promising insights into the differential expression of these miRNAs in early and advanced disease stages. However, several aspects need clarification or improvement before the manuscript can be considered for publication.

However, several critical issues need to be addressed before it can be considered for publication.

My comments are described as follows:

Comments:

  1. No multivariate analysis was presented to assess the independence of miR-31-3p or miR-196a-5p as biomarkers when adjusted for confounding variables including age, sex, smoking, alcohol. This analysis would strengthen the claim of their diagnostic utility.The scale bars are missing in the figures of wound healing assays and invasion assays.
  2. The study includes a relatively limited cohort comprising 50 patients and 20 healthy controls, which may restrict the robustness of the findings for biomarker validation. Although the reported differences in miRNA expression are statistically significant, the authors should provide a discussion on the statistical power of the study and address how the sample size may influence the reliability and generalizability of the results to broader clinical populations.
  3. Although the discussion outlines several pathways potentially regulated by miR-31-3p and miR-196a-5p involved in LATS2, FIH, and HOX gene families, the study lacks functional validation through experimental approaches such as in vitro or in vivo assays. The authors are encouraged to acknowledge this as a limitation and to emphasize the need for future mechanistic studies to elucidate the biological roles of these miRNAs in LSCC pathogenesis.
  4. The figures, including boxplots and ROC curves, effectively illustrate key findings; however, their visual quality could be enhanced. The authors should improve the resolution and ensure that all axes are clearly labeled, including appropriate units, scales, and definitions, to facilitate accurate interpretation and enhance overall clarity.
  5. The manuscript provides a detailed discussion of the biological functions of miR-196a-5p, however its relatively limited discriminatory performance in the current study is not adequately addressed. The authors should clarify whether miR-196a-5p holds standalone diagnostic value or if it is better suited as part of a combined biomarker panel alongside miR-31-3p or other markers.

Author Response

Dear Authors,

The manuscript explores the diagnostic potential of two circulating microRNAs miR-31-3p and miR-196a-5p as liquid biopsy markers in HPV-negative laryngeal squamous cell carcinoma (LSCC). The study is timely and relevant, given the ongoing search for non-invasive biomarkers in head and neck cancers. The methodology is sound, and the results provide promising insights into the differential expression of these miRNAs in early and advanced disease stages. However, several aspects need clarification or improvement before the manuscript can be considered for publication.

However, several critical issues need to be addressed before it can be considered for publication.

My comments are described as follows:

Comments:

  1. No multivariate analysis was presented to assess the independence of miR-31-3p or miR-196a-5p as biomarkers when adjusted for confounding variables including age, sex, smoking, alcohol. This analysis would strengthen the claim of their diagnostic utility.The scale bars are missing in the figures of wound healing assays and invasion assays.

Response 1: Thank you for your comment. The multivariate analysis would improve the knowledge about miR-31-3p or miR-196a -5p. In our feature study, we will include it. The scale bars of the Figures 1 and 2 are improved. I do not understand the phrases “in the figures of wound healing assays and invasion assays.”. I am sorry about that.

  1. The study includes a relatively limited cohort comprising 50 patients and 20 healthy controls, which may restrict the robustness of the findings for biomarker validation. Although the reported differences in miRNA expression are statistically significant, the authors should provide a discussion on the statistical power of the study and address how the sample size may influence the reliability and generalizability of the results to broader clinical populations.

Response 2: Thank you for your comment. The limitations of the study are improved and added in the Discussion.

  1. Although the discussion outlines several pathways potentially regulated by miR-31-3p and miR-196a-5p involved in LATS2, FIH, and HOX gene families, the study lacks functional validation through experimental approaches such as in vitro or in vivo assays. The authors are encouraged to acknowledge this as a limitation and to emphasize the need for future mechanistic studies to elucidate the biological roles of these miRNAs in LSCC pathogenesis.

Response 3: Thank you for your comment. The limitations of the study are improved and added in the Discussion.

  1. The figures, including boxplots and ROC curves, effectively illustrate key findings; however, their visual quality could be enhanced. The authors should improve the resolution and ensure that all axes are clearly labeled, including appropriate units, scales, and definitions, to facilitate accurate interpretation and enhance overall clarity.

Response 4: Agree. The Tables and Figures are improved.

  1. The manuscript provides a detailed discussion of the biological functions of miR-196a-5p, however its relatively limited discriminatory performance in the current study is not adequately addressed. The authors should clarify whether miR-196a-5p holds standalone diagnostic value or if it is better suited as part of a combined biomarker panel alongside miR-31-3p or other markers.

Response 5: Thank you for your comment. The manuscript is improved. Please check Discussion lines: 238-240 and conclusion.

Round 2

Reviewer 1 Report

Comments and Suggestions for Authors

Authors have corrected paper according to to my comments. I have no addtional ones.

Author Response

Dear Reviewer,

Thank you very much!

Your sincerely,

Silva Kyurkchiyan

Reviewer 2 Report

Comments and Suggestions for Authors

The revision is not well-improved. On the contrary, miR-196a-5p is taken out of the conclusions. Now, the significance of this study is poor.

In my question 3. I mentioned that “Table 1 showed miR-196-5p has no significant association with LSCC’s tumor stage, nodal stage, and G stage. Figure 3 and Line 250 showed that miR-196a-5p's ability to distinguish early from advanced LSCC was not statistically significant (p=0.255). Therefore, these results cannot support the conclusion of miR-196a-5p has potential as LSCC screening markers.” In responses, authors fail to explain why miR-196-5p is not significant association with LSCC’s tumor stage, nodal stage, and G stage, and is not able to distinguish early from advanced LSCC. They simply take miR-196-5p out of conclusion. Paradoxically, in the title authors still emphasize miR-196a-5p as Liquid Biopsy Markers.

In my question 4, I mentioned that “Figure 4 legend: “Spearman correlation between circulating miR-31-3p and miR-196a-5p showed positive co-expression with correlation coefficient rs=0.508 (p=0.001). The fitted line was added to models the relationship between both variables.” However, Figure 4 didn’t show good correlation between miR-31-3p and miR-196-5p. Please provide the original data of figure 4.” The related claims and conclusion in this revision are still wrong. In addition, authors didn’t provide original data. Authors said the figure is original data. However, the figure is not original data. The numbers of their relative expression levels are original data.

In question 2. I asked for the detail method of their calculation of relative quantification (RQ). However, authors fail to do so. This method may be simple, but is a relative quantification method. Therefore, how to set up control is very important.  

“The mean RQ of miR-31-3p in the control group was 1.07 ± SD 0.71 (median = 0.79),” It is wrong. Mean is not SD.

In question 1, I mentioned that “Figure 1. The number of cases in each group should be added in legend. The results of statistical analysis should be added in the figure. Please provide detail statistical method about this figure.” However, authors just said “SPSS software was used to be created the boxplots.”. They fail to provide correct statistical method.

Author Response

Dear Editor 2,

Thank you very much for detailed manuscript revision and precise recommendations.

Please, find my comment below each of your question. The answers are marked with red font bolded color.

The revision is not well-improved. On the contrary, miR-196a-5p is taken out of the conclusions. Now, the significance of this study is poor.

Answer: Thank you for your comment. Please find the corrected conclusion below.

According to the revision comment, miR-196a-5p showed limited clinical significance, and I have removed this miRNA from the conclusion to place greater emphasis on miR-31-3p. However, based on my analysis and a review of the literature, miR-196a-5p remains of considerable interest. We were not able to detect a significant difference in miR-196a-5p expression between early- and advanced-stage laryngeal cancer samples. Nevertheless, miR-196a-5p expression in control groups was within the normal range, whereas it was already elevated in early-stage laryngeal cancer. This finding suggests that miR-196a-5p could play a key role in very early stages of malignancy, during the development of cancer lesions, when it is overexpressed.

In my question 3. I mentioned that “Table 1 showed miR-196-5p has no significant association with LSCC’s tumor stage, nodal stage, and G stage. Figure 3 and Line 250 showed that miR-196a-5p's ability to distinguish early from advanced LSCC was not statistically significant (p=0.255). Therefore, these results cannot support the conclusion of miR-196a-5p has potential as LSCC screening markers.” In responses, authors fail to explain why miR-196-5p is not significant association with LSCC’s tumor stage, nodal stage, and G stage, and is not able to distinguish early from advanced LSCC. They simply take miR-196-5p out of conclusion. Paradoxically, in the title authors still emphasize miR-196a-5p as Liquid Biopsy Markers.

Answer: Thank you for your comment and question.

In our analyzed cohort of patients, miR-196a-5p did not reach significance in distinguishing advanced from early laryngeal cancer. However, miR-196a-5p showed promising results in differentiating early LSCC from healthy controls. ROC analysis indicated that miR-196a-5p has potential utility as a screening marker for detecting early-stage LSCC in clinical practice, which is crucial for improving outcomes in laryngeal cancer. Laryngeal cancer is often diagnosed at advanced stages, and recurrence rates are substantial. The identification and implementation of a screening biomarker could improve early detection of laryngeal malignancy, enabling treatment options that may spare the organ. We believe that miR-196a-5p has biomarker potential. Our future projects will explore the use of miR-31-3p and miR-196a-5p (alone and in combination) for monitoring cancer recurrence. These results will inform future conclusions.

In my question 4, I mentioned that “Figure 4 legend: “Spearman correlation between circulating miR-31-3p and miR-196a-5p showed positive co-expression with correlation coefficient rs=0.508 (p=0.001). The fitted line was added to models the relationship between both variables.” However, Figure 4 didn’t show good correlation between miR-31-3p and miR-196-5p. Please provide the original data of figure 4.” The related claims and conclusion in this revision are still wrong. In addition, authors didn’t provide original data. Authors said the figure is original data. However, the figure is not original data. The numbers of their relative expression levels are original data.

Answer: Thank you for your comment and question.

Unfortunately, I do not understand why you is suspicion about the data. The level of correlation coefficient is moderately to strong. I do not understand your meaning of “good correlation”. The correlation is what it is. The meaning of correlation guides us how strong is the relationship between both miRNAs. In the research and statistical fields, there is not such term “good” or “bad” correlation. The correlation is significant or not significant. In our study correlation between both miRNAs is significant and correlation coefficient is 0.508. This means that more than half of the miR-31-3p and miR-196a-5p molecules are expressed monotonically, and the elevated levels of miR-31-3p correspond with 196a-5p elevated levels. Here the point in Spearman correlation is that the tendence of increasing of both molecules is not obligatory related to their direct interrelationship. Perhaps other factors influence their overexpression. In the correlation analysis are used the RQ data, which is explained in the figure. What exactly data do you mean as “original data”. Our RQ values are not fake, they are real and properly calculated. Moreover, our team works on laryngeal cancer for such a long time and our aim is to try to develop a potential marker for the use of laryngeal cancer screening or management. We do not use fake data. Please, provide me in details what do you mean with “original data”. Already few times you mentioned that. You could check our team on Google. Prof. Todor Popov is the best ENT specialist in Bulgaria, and Prof. Radka Kaneva is the Head of Molecular Center Laboratory, which is main center and place for research including student’s education. Prof. Diana Popova is also a great ENT specialist. She introduced and developed the use of hearing aids for the first time in Bulgaria. She was the Head of the ENT department in Medical University-Sofia, and her research contribution is well-known. We are not cheaters.

If you do not believe in RQ values, please provide me an email and in confidential agreement and disclosure I will send you all data that I have.

In question 2. I asked for the detail method of their calculation of relative quantification (RQ). However, authors fail to do so. This method may be simple, but is a relative quantification method. Therefore, how to set up control is very important. 

Answer: Thank you for your question. I am sorry, that I failed to explain the detailed relative quantification steps.

First, we choose the best endogenous control, which will be used to calculate the next steps.

The endogenous control must be with stable expression in control and target groups.

During our previous project, we have tested few endogenous controls, including: RNU6, RNU44, RNU48, and miR-16 (spike-in control). The three RNU6, RNU44 and RNU48 endogenous controls were chosen from the list of most common endogenous controls. In some studies, included in the kits miR-16 spike-in control is used from the researchers as endogenous control. However, we have tested the miR-16, but it gives only information about the miRNA extraction precision. Our team find it as not the best choice as endogenous control.

During the test analysis of RNU6, RNU44 and RNU48, RNU6 showed most stable expression between control and laryngeal cancer samples. And it was used in our following projects. In the current project RNU6 also showed stable Ct expression levels.

The next step was to determine the Ct levels. We run qPCR on both of our genes of interest and the endogenous control for all samples (controls and patients) on the real-time qPCR machine. The Ct (threshold cycle) values represent the PCR cycle number at which the fluorescence signal exceeds a threshold. We set 40 PCR cycles.

We subtracted the Ct value of the endogenous control from the Ct value of the gene of interest (miR-31-3p and miR-196a-5p) for each sample. This step was used to normalize the data to the endogenous control.

We set the calibrator reference for each of both target miRNAs. The calibrator reference was set as we calculated the average of delta Ct (dCt) of all Ct values for miR-31-3p and miR-196a-5p. We subtracted the Ct of the target RNA from Ct of the endogenous control, as we obtained delta Ct (dCt) for each analyzed target miRNA per control sample. The next step was to average all miR-31-3p or miR196a-5p dCts values, as we obtained average miR-31-3p dCt as reference value for calculations of the target miR-31-3p in the patients’ samples, and miR-196a-5p dCt as reference value for calculations of the target miR-196a-5p in the patients’ samples.

            Then we calculated the delta Ct values for miR-31-3p and miR-196a-5p in the target patient’s group. We subtracted Ct values of the target miR-31-3p or miR-196a-5p from the Ct values of the endogenous control. This represents delta Ct (dCt) values for each of both miR-31-3p and miR-196a-5p in all patients’ samples.

Following that, we calculated delta delta Ct (ddCt) values, as we subtracted dCt of the target miRNA (miR-31-3p or miR-196a-5p, respectively) in the patients’ sample from the average dCt value (the reference value for mR-31-3p or miR-196a-5p) used as reference value.

            Obtained ddCt values was used to be calculated the relative quantification or relative expression of both target miRNAs in the patient’s group. The mathematical equation that was used for the RQ calculation was: RQ = 2^(-ΔΔCt).

            Additionally, we calculated RQ values for the expression of miR-31-3p and miR-196a-5p in the controls. For the aim we have subtracted the dCt of miR-31-3p or miR-196a-5p from the reference value (the average dCt for each miRNA in the control samples). Then the calculated ddCt value was used in RQ calculation: RQ = 2^(-ΔΔCt).

The last step was used in order to be presented the data in clearer format for the researcher and people who will read the manuscript. We believe, that this presentation gives us more precise view of the data.

In the manuscript was not written the RQ method in details, because as you can see it is a long writing. Instead, we put a reference of a paper, where all information about the method in described.

We used a positive and negative control to verify the correct amplification producing.

“The mean RQ of miR-31-3p in the control group was 1.07 ± SD 0.71 (median = 0.79),” It is wrong. Mean is not SD.

Answer: Thank you for your comment. Yes, you are right mean is not SD. I wrote mean ± SD, due to that the SD values give information about the spread or dispersion of data points around that mean. The ± SD was added after mean between lines 229 – 237.

In question 1, I mentioned that “Figure 1. The number of cases in each group should be added in legend. The results of statistical analysis should be added in the figure. Please provide detail statistical method about this figure.” However, authors just said “SPSS software was used to be created the boxplots.”. They fail to provide correct statistical method.

Answer: Thank you for your comment and question.

The data included in the Figure 1 was calculated by non-parametrical method Kruskal-Wallis test. I used this test, due to that the data fails the test for normal distribution. In this case the most appropriate statistical tests are non-parametric. Due to that part of the readers of the studies are more oriented with mean ± SD we included the corresponding values as wee, and in brackets are written the medians of each group. Because the Kruskal-Wallis method uses the medians for statistical analysis. We would like to be as much as transparent with our study.

The sentence “Kruskal-Wallis statistical test was used to compare the three groups, including the controls, early stage LSCC and advanced sage LSCC groups.” in the lines 237-238.

In the end of the text next to Figure 1 was included in brackets Kruskal-Wallis statistical method.

If you need additional information, please let me know!

Your sincerely,

Silva Kyurkchiyan

Reviewer 3 Report

Comments and Suggestions for Authors

I have no more comments on the manuscript.

Author Response

(The authors gave the same response as above.)

Reviewer 4 Report

Comments and Suggestions for Authors

Dear Authors,

Thank you for your revision. I have no more questions.

Author Response

(The authors gave the same response as above.)
